# Increasing the Efficiency of Canola and Soybean GMO Detection and Quantification Using Multiplex Droplet Digital PCR

**DOI:** 10.3390/biology11020201

**Published:** 2022-01-27

**Authors:** Tigst Demeke, Sung-Jong Lee, Monika Eng

**Affiliations:** Grain Research Laboratory, Canadian Grain Commission, Winnipeg, MB R3C 3G8, Canada; sung-jong.lee@grainscanada.gc.ca (S.-J.L.); monika.eng@grainscanada.gc.ca (M.E.)

**Keywords:** droplet digital PCR (ddPCR), multiplex, GM event, canola, soybean, detection, duplex, triplex, tetraplex

## Abstract

**Simple Summary:**

Digital PCR (dPCR) technology has been used for absolute quantification of genetically modified (GM) events. Duplex dPCR consisting of a target gene and a reference gene is mostly used for absolute quantification of GM events. We investigated the feasibility of absolute quantification of two, three, and four GM canola and soybean events at the same time using the QX200 Droplet Digital PCR (ddPCR) system. Adjustments of the probe concentrations and labels for some of the assays were needed for successful multiplex ddPCR. Absolute quantification of GM canola and soybean events was achieved for duplex, triplex, and tetraplex ddPCR at 0.1%, 1%, and 5% concentrations.

**Abstract:**

The number of genetically modified (GM) events for canola, maize, and soybean has been steadily increasing. Real-time PCR is widely used for the detection and quantification of individual GM events. Digital PCR (dPCR) has also been used for absolute quantification of GM events. A duplex dPCR assay consisting of one reference gene and one GM event has been carried out in most cases. The detection of more than one GM event in a single assay will increase the efficiency of dPCR. The feasibility of detection and quantification of two, three, and four GM canola and soybean events at the same time was investigated at 0.1%, 1%, and 5% levels using the QX200 Droplet Digital PCR (ddPCR) system. The reference gene assay was carried out on the same plate but in different wells. For some of the assays, optimization of the probe concentrations and labels was needed for successful ddPCR. Results close to the expected result were achieved for duplex, triplex, and tetraplex ddPCR assays for GM canola events. Similar ddPCR results were also achieved for some GM soybean events with some exceptions. Overall, absolute quantification of up to four GM events at the same time improves the efficiency of GM detection.

## 1. Introduction

The number of genetically modified (GM) events has been increasing since the start of commercialization of GM crops in the mid-1990s [1]. Many countries have a regulatory approval process for new GM events and effective detection methods are required for monitoring [2,3,4]. The enforcement of labeling laws for GM grains and derived products also necessitates effective detection methods. The main reason for enforcement is because of consumers’ rights to know what is in their food. New breeding technologies, such as CRISPR-Cas9, are also being used for crop improvement and information about genetic modification is required if there is a need for monitoring [5]. Real-time qualitative and quantitative PCR assays have been widely used for the detection and quantitative analysis of GM events [6,7,8]. However, the real-time PCR assay relies on the availability of certified reference materials and use of appropriate standard curves. Two calibration curves (for transgene and endogene) are used for real-time PCR to calculate the amount of target DNA present in the unknown sample [9]. The minimum required performance limit (MRPL), which is the lowest level of GM material required for the validation of quantitative methods, has been set by European Union Reference Laboratory at 0.1% [10].

At the moment, digital PCR (dPCR) is being used by many laboratories for the detection and quantification of GM events [11,12]. Digital PCR is more convenient than real-time PCR since it does not require a standard curve. Collaborative studies have shown high reproducibility of dPCR assays [13,14]. An inter-laboratory dPCR study comprising seven independent laboratories showed less than 4.5% reproducibility relative standard deviation for a high GC-rich reference material [13]. dPCR assays have also been reported to be less sensitive to inhibitors in the DNA samples compared with real-time quantitative PCR assays [15,16,17]. However, ethanol has been reported to affect the stability of droplets for dPCR [15].

Digital PCR is used for absolute quantification of target nucleic acids in a GM sample. Droplet-based and chip-based dPCR instruments are used for the detection of GM events and other applications [18]. The PCR solution is partitioned into different sub-reactions or droplets. The regular PCR instrument is used to amplify the individual sub-reactions and Poisson statistics are used to determine the concentration of the target sequence [19]. The use of primers and probes for one target and one reference gene in the same PCR (duplex dPCR) has been routinely used for the detection and quantification of GM events [20,21,22]. However, the detection and quantification of one GM event at a time is not efficient. The Bio-Rad QX200 has two fluorescence filters, which is suitable for performing duplex droplet digital PCR (ddPCR). Optimization of the probe concentrations and ratios was conducted for absolute quantification of more than two targets for spinal muscular atrophy [23]. The detection of four GM maize events at the same time using ddPCR has been reported [24]. Two 4-plex assays consisting of 4 GM events for 1 set and 3 GM events and an endogene for the second set were used for the multiplex ddPCR maize assay. The basic principles of multiplexing with dPCR have also been documented [25]. 

In general, the wide applicability of multiplex dPCR for the detection of GM events has not been investigated. A successful multiplex dPCR assay for GM events will enhance the efficiency of detection and save resources. The objective of this study was to assess the feasibility of the quantification of two, three, and four GM canola and soybean events at the same time using the QX200 ddPCR system. Our plan was to investigate whether the multiplex ddPCR works for two GM events initially and then proceed to three and four GM events. The adoption rate for biotech canola and soybean in Canada was reported to be 92.5% in 2018 [26]. The development of efficient detection methods for discontinued and new GM canola and soybean events is important for monitoring. 

## 2. Materials and Methods

### 2.1. Sources of Seeds and Reference Materials

The seeds and reference materials used for the canola ddPCR assay were Legend (non-GM canola, certified seed), Conquest (breeder seed-GT73 GM event–Glyphosate herbicide tolerance), Innovator (breeder seed-HCN92 GM event–Glufosinate herbicide tolerance), Armor BX (breeder seed-OXY235 GM event–Oxynil herbicide tolerance), and AOCS 1011-A (certified reference material, CRM-MON88302 GM event–Glyphosate herbicide tolerance). The seeds of Legend, Conquest, and Innovator were obtained from Oilseeds Program of the Grain Research Laboratory.

The seeds and reference materials used for soybean ddPCR assay were Colby (non-GM soybean variety), PS 2295 LL (certified seed-A2704 GM event–Glufosinate herbicide tolerance), ERM-BF426d (CRM-DP305423 GM event-Sulfonylurea herbicide, modified oil/fatty acid), AOCS-0906B2 (CRM-MON89788 GM event–Glyphosate herbicide tolerance), and ERM-BF437-B (CRM-DAS81419 GM event–Glufosinate herbicide tolerance, Lepidopteran insect resistance). Certified seeds of Colby were obtained from WG Thompson & Sons (Granton, ON, Canada) and certified seeds of PS 2295 LL were obtained from Bayer Crop Science. All the canola and soybean varieties were grown in Canada.

### 2.2. DNA Extraction and Quantification

DNA was extracted using a DNeasy mericon Food kit (Qiagen Life Sciences, LLC, Louisville, KY, USA). Agarose gel-electrophoresis (1.2%) was used to check the quality of DNA. 1X TAE (Tris-Acetate-EDTA) buffer containing 0.44 mM GelRed dye (Biotium, Cedarlane Laboratories, Hornby, Ontario—sourced from Fremont, CA, USA) was used for gel-electrophoresis. The Pico Green^®^ assay (Molecular Probes, Eugene, OR, USA) was used to measure the amount of DNA and λ DNA was used to generate a calibration curve. The Pico Green^®^ dsDNA reagent was added to diluted DNA in a 96-well plate. A SpectraMax M5 Microplate Reader (Molecular Devices, Toronto, ON, Canada—sourced from San Jose, CA, USA) was used to measure the fluorescence in duplicates. In total, 100 ng of DNA were used for all ddPCR experiments. Then, 0.1%, 1%, and 5% GM samples were prepared by mixing non-GM and GM DNA. A control without GM DNA was included for the experiments. A DNA mixture consisting of the different GM events was prepared for duplex, triplex, and tetraplex ddPCR. An example of how the DNA mixture was prepared is provided in the Appendix A. 

### 2.3. Droplet Digital PCR

A Bio-Rad QX200 instrument was used for ddPCR as described by Demeke et al. [27]. The event-specific primer and probe DNA sequences used for the experiments are shown in Table 1. The canola and soybean GM events were chosen based on in-house validation work done and successful participation in proficiency test programs. The primer and probe DNA sequences for the single copy reference genes used were FatA(A) for canola [28] and Lectin for soybean [29]. Canola and soybean GM events were assessed for the multiplex ddPCR. The reference gene assay was run with the ddPCR on the same plate separately so that more target genes could be assessed for the multiplex ddPCR. In total, 3 replications of the reference gene assay (FatA(A) or Lectin) were carried out for each of the 3 GM concentrations determined (0.1%, 1%, and 5%). The average number of reference droplets was used for absolute quantification of each GM event. The concentrations of the primers and probes used for duplex, triplex, and tetraplex ddPCR are shown in Appendix A, respectively.

QuantaSoft 1.7.4.0917 and automatic threshold were used for conducting the ddPCR assays. QX Manager Software (version 1.2) was used for analyzing tetraplex ddPCR assays. 

## 3. Results and Discussion

### 3.1. Optimization of the Probe Concentrations

The QX200 ddPCR system has two fluorescence filters and it is a challenge to successfully run multiplex ddPCR involving more than two targets. Optimization of the probe concentrations is required for successful multiplex ddPCR consisting of more than two targets [24]. The primer and probe concentrations used for the ddPCR assays in this study were based on published information for real-time event-specific PCR assays [30,31] and in-house validation. The forward and reverse primer concentrations for 7 of the 8 target genes for ddPCR were kept at 0.4 μM. For the DP305423 soybean event, forward and reverse primer concentrations of 0.8 μM and 0.5 μM were used, respectively [32]. For probe optimization, all four GM events were run individually using FAM and HEX probe dyes in triplicates to determine the signal intensity and establish the cluster location. Based on the results, the probes for the GM events to be labeled with either FAM or HEX were determined. Then, all four GM events were run together to assess the pattern and separation of clusters. After that, the probe concentrations were adjusted in order to obtain clear separation of the clusters. The concentrations of primers and probes and the probe labels used for duplex, triplex, and tetraplex ddPCR are shown in Appendix A. Increased probe concentrations were used for some of the events for triplex and tetraplex ddPCR compared to duplex ddPCR. Mixtures of equal amounts of FAM- and HEX-labeled probes were used for MON88302 and DP305423 GM events in order to obtain the third position for triplex ddPCR. Distinct separation of all positive droplet clusters was not obtained before optimization of the probe concentrations (Figure 1). 

### 3.2. Detection of Two GM Events (Duplex ddPCR)

Three pairs of GM canola events were assessed for the detection of 0.1%, 1%, and 5% spiked samples (Table 2). Close to the expected results were achieved for the HCN92 and GT73, HCN92 and MON88302, and GT73 and MON88302 duplex ddPCR assays. Three pairs of GM soybean events (A2704 and DP305423, A2704 and MON89788, and DP305423 and MON89788) were also assessed for three concentrations of spiked DNA samples (Table 2). For the soybean duplex ddPCR assay, some of the results for the 0.1% and 1% samples were higher than the expected values. The reference gene assay carried out in different wells of the same plate to that of the duplex ddPCR assay was suitable for absolute quantification. An example of the distribution of droplets that formed for duplex canola ddPCR is depicted in Figure 2. Quantifying two GM events at the same time is an improvement of the routine duplex ddPCR assay, which consists of one GM event and the reference gene. 

### 3.3. Detection of Three GM Events (Triplex ddPCR)

Spiked DNA samples containing three GM events were assessed at three concentrations (Table 3). The first triplex ddPCR canola assay consisted of HCN92, GT73, and MON88302 events while the second triplex ddPCR canola assay consisted of OXY235, GT73, and MON88302 events. Close to the expected values were achieved for the triplex canola ddPCR assays. An example of the distribution of droplets for the triplex ddPCR canola assay is shown in Figure 3. Improved resolution of the different droplets was observed at the highest concentration (5%). The seven positive clusters and the negative cluster were well separated. The triplex ddPCR soybean assay consisted of A2704, DP305423, and MON89788 for one set and A2704, DAS81419, and MON89788 for the second set (Table 3). Close to the expected results were achieved, with the exception of DP305423 at 0.1%. 

### 3.4. Detection of Four GM Events (Tetraplex ddPCR)

Careful optimization of the probe concentrations was required for the tetraplex ddPCR assays. Different amounts of probe concentrations were tested in order to obtain clear separation of the different clusters. The tetraplex ddPCR assay results for four GM canola events (HCN92, GT73, OXY235, and MON88302) are shown in Table 4. Close to the expected results were achieved for the 0.1%, 1%, and 5% GM canola samples. The tetraplex ddPCR for the GM soybean assay consisted of A2704, DP305423, DAS81419, and MON89788 events. Close to the expected results were achieved for three of the four GM events (Table 4). For the DP305423 GM soybean event, higher than expected results were obtained. The results for this event were also higher for the duplex and triplex ddPCR. A two-dimensional view of the separation of droplets for the 15 possible combinations for the tetraplex ddPCR soybean assay is provided in Figure 4. There was a clear separation of the negative and the 15 positive droplet clusters. Clear separation of droplet clusters was not obtained before optimization of probe concentrations (Figure 1). It was relatively easy and faster to identify the clusters with the QX Manager software. 

Overall, successful tetraplex ddPCR assays were developed for absolute quantification of GM canola and soybean events at 0.1%, 1%, and 5% levels. Amplification was achieved for all samples at 0.1%, which is the minimum required performance limit for the validation of quantitative methods [10]. The duplex, triplex, and tetraplex ddPCR canola and soybean assays described in this study will be applicable for other GM events. Optimization of the probe concentrations may be needed for cluster separation and successful multiplex ddPCR assay of other GM events. 

## 4. Conclusions

In most of the reports of GM detection and quantification using dPCR, a duplex assay consisting of a GM event and a reference gene was used at the same time. However, it is important to increase the efficiency of GM event detection using multiplex ddPCR. The feasibility of the absolute quantification of two, three, and four GM events at the same time was investigated in this study. Optimization of the probe concentrations and labels was necessary for some of the assays in order to obtain successful multiplex ddPCR results. Absolute quantification of two, three, and four GM canola and soybean events was achieved using the QX200 ddPCR instrument at 0.1%, 1%, and 5% levels. The developed multiplex ddPCR assays will help to enhance the efficiency of GM detection and quantification for canola and soybean events. The duplex, triplex, and tetraplex ddPCR canola and soybean assays described in this study are also applicable to the detection and quantification of other GM events. 

## Figures and Tables

**Figure 1 biology-11-00201-f001:**
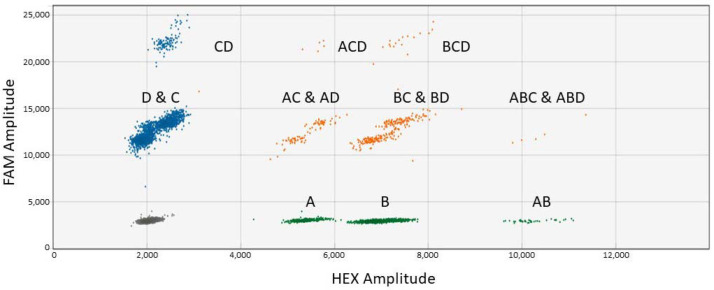
Droplet clusters generated for the non-optimized soybean tetraplex ddPCR assay (two-dimensional view). A = DP305423 event (HEX); B = DAS81419 event (HEX); C = MON89788 event (FAM); and D = A2704 event (FAM). The dark cluster represents negative droplets.

**Figure 2 biology-11-00201-f002:**
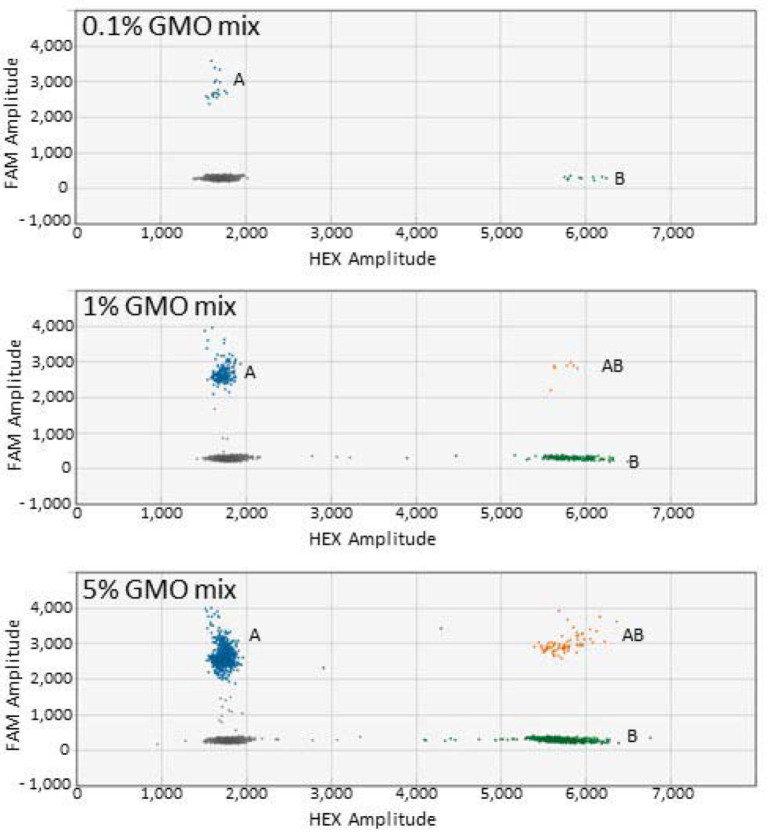
Droplet clusters generated for the duplex canola ddPCR assay. A = HCN92 (FAM); B = GT73 (HEX); and AB = Contains both HCN92 and GT73 droplets. The dark cluster represents negative droplets.

**Figure 3 biology-11-00201-f003:**
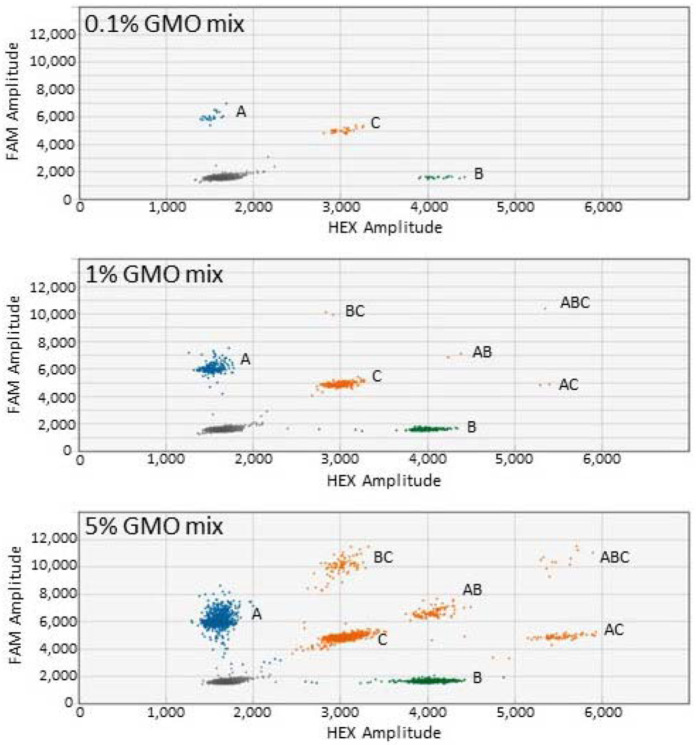
Droplet clusters generated for the canola triplex ddPCR assay (two-dimensional view). A = HCN92 (FAM); B = GT73 (HEX); C = MON88302 (FAM and HEX). The dark cluster represents negative droplets.

**Figure 4 biology-11-00201-f004:**
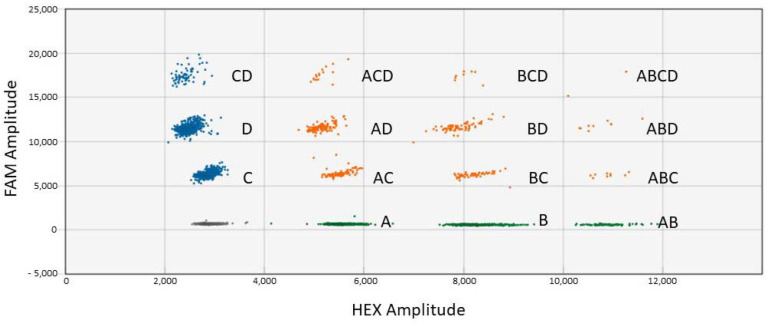
Droplet clusters generated for the soybean tetraplex ddPCR assay (two-dimensional view). A = DP305423 event (HEX); B = DAS81419 event (HEX); C = MON89788 event (FAM); and D = A2704 event (FAM). The dark cluster represents negative droplets.

**Table 1 biology-11-00201-t001:** Primer and probe DNA sequences used for absolute quantification of GM canola and soybean events.

Crop	Event Name	Primer/Probe Name	DNA Sequences (5′ to 3′)	Amplicon Length (bp)
CANOLA	GT73	RT73-1	CCA TAT TGA CCA TCA TAC TCA TTG CT	108
		RT73-2	GCT TAT ACG AAG GCA AGA AAA GGA	
		RT73 (P)	TTC CCG GAC ATG AAG ATC ATC CTC CTT-BHQ1	
	OXY235	Oxy RG	GAT AGA TGG TGG TGT GAG TCT TGT	124
		OXY RV	CCT AAC TTT TGG TGT GAT GAT GCT	
		Oxy RP	TGC CAT CAG CTG ACA CGC CGT GC-BHQ1	
	HCN92	MDB685	GTT GCG GTT CTG TCA GTT CC	95
		KVM180	CGA CCG GCG CTG ATA TAT GA	
		TM029	TCC CGC GTC ATC GGC GG-BHQ1	
	MON88302	88302QF	TCC TTG AAC CTT ATT TTA TAG TGC ACA	101
		88302QR	TCA GAT TGT CGT TTC CCG CCT TCA	
		88302QP	TAG TCA TCA TGT TGT ACC ACT TCA AAC ACT-BHQ1	
	FatA(A)	09-0-2824-F	ACA GAT GAA GTT CGG GAC GAG TAC	84
		09-0-2825-R	CAG GTT GAG ATC CAC ATG CTT AAA TAT	
		09-QP-87-P	AAG AAG AAT CAT CAT GCT TC-BHQ1	
SOYBEAN	DP305423	DP305 F1	CGT GTT CTC TTT TTG GCT AGC	93
		DP305 R5	GTG ACC AAT GAA TAC ATA ACA CAA ACT A	
		DP305423 P	TGA CAC AAA TGA TTT TCA TAC AAA AGT CGA GA-BHQ1	
	A2704	KVM175	GCA AAA AAG CGG TTA GCT CCT	74
		SMO 001	ATT CAG GCT GCG CAA CTG TT	
		TM021	CGG TCC TCC GAT CGC CCT TCC-BHQ1	
	MON89788	MON89788 F	TCC CGC TCT AGC GCT TCA AT	139
		MON89788 R	TCG AGC AGG ACC TGC AGA A	
		MON89788 P	CTG AAG GCG GGA AAC GAC AAT CTG-BHQ1	
	DAS81419	DAS81419 F2	TCT AGC TAT ATT TAG CAC TTG ATA TTC AT	105
		DAS81419 R1	GCT TCA AGA TCC CAA CTT GCG	
		DAS81419 P3	ATC AAC AGG CAC CGA TGC GCA CCG-BHQ1	
	Lectin1	lec-F	CCA GCT TCG CCG CTT CCT TC	74
		lec-R	GAA GGC AAG CCC ATC TGC AAG CC	
		lec-P	CTT CAC CTT CTA TGC CCC TGA CAC-BHQ1	

Reporter dyes used are shown in the Appendix A. FatA(A) and Lectin1 are the reference gene primers and probes used for canola and soybean, respectively.

**Table 2 biology-11-00201-t002:** Experimental results for duplex ddPCR assays for GM canola and soybean events.

Crop	Duplex Events	Event	Experimental Results ^a^		
			0.1%	1%	5%
Canola	HCN92 and GT73	HCN92	0.08 ± 0.01	1.12 ± 0.07	5.20 ± 0.48
		GT73	0.08 ± 0.03	1.00 ± 0.12	5.30 ± 0.40
	HCN92 and MON88302	HCN92	0.07 ± 0.01	0.94 ± 0.08	4.53 ± 0.42
		MON88302	0.10 ± 0.01	1.06 ± 0.06	5.10 ± 0.22
	GT73 and MON88302	GT73	0.08 ± 0.03	0.94 ± 0.10	4.55 ± 0.25
		MON88302	0.08 ± 0.02	0.93 ± 0.07	5.10 ± 0.25
Soybean	A2704 and DP305423	A2704	0.16 ± 0.02	1.26 ± 0.10	4.69 ± 0.68
		DP305423	0.18 ± 0.06	1.56 ± 0.07	5.57 ± 0.65
	A2704 and MON89788	A2704	0.13 ± 0.03	1.16 ± 0.08	4.49 ± 0.65
		MON89788	0.15 ± 0.02	1.35 ± 0.05	4.93 ± 0.69
	DP305423 and MON89788	DP305423	0.15 ± 0.03	1.50 ± 0.02	5.65 ± 0.24
		MON89788	0.12 ± 0.00	1.33 ± 0.03	4.83 ± 0.14

^a^ Average of three replications ± standard deviation.

**Table 3 biology-11-00201-t003:** Experimental results for triplex ddPCR assays for GM canola and soybean events.

Crop	Triplex Events	Event	Experimental Results ^a^		
			0.1%	1%	5%
Canola	HCN92, GT73,	HCN92	0.10 ± 0.03	1.09 ± 0.08	5.15 ± 0.30
	and MON88302	GT73	0.12 ± 0.01	1.07 ± 0.05	4.99 ± 0.12
		MON88302	0.11 ± 0.01	1.05 ± 0.03	5.09 ± 0.32
Canola	OXY235, GT73, and	OXY235	0.13 ± 0.01	1.38 ± 0.07	6.27 ± 0.19
	MON88302	GT73	0.12 ± 0.01	1.17 ± 0.02	5.31 ± 0.06
		MON88302	0.12 ± 0.01	1.12 ± 0.05	5.22 ± 0.18
Soybean	A2704, DP305423,	A2704	0.11 ± 0.03	0.94 ± 0.09	3.91 ± 0.10
	and MON89788	DP305423	0.17 ± 0.05	1.22 ± 0.04	5.21 ± 0.32
		MON89788	0.12 ± 0.02	1.01 ± 0.14	4.51 ± 0.33
Soybean	A2704, DAS81419,	A2704	0.10 ± 0.05	1.03 ± 0.04	4.90 ± 0.17
	and MON89788	DAS81419	0.14 ± 0.03	1.35 ± 0.06	6.20 ± 0.17
		MON89788	0.10 ± 0.01	1.15 ± 0.05	5.41 ± 0.22

^a^ Average of three replications ± standard deviation.

**Table 4 biology-11-00201-t004:** Experimental results for the tetraplex ddPCR assays for GM canola and soybean events.

Crop	Tetraplex Events	Event	Experimental Results ^a^		
			0.1%	1%	5%
Canola	GT73, MON88302,	GT73	0.11 ± 0.02	1.07 ± 0.05	5.01 ± 0.19
	OXY235, and HCN92	MON88302	0.12 ± 0.02	1.10 ± 0.04	5.07 ± 0.18
		OXY235	0.12 ± 0.01	1.26 ± 0.10	5.75 ± 0.12
		HCN92	0.10 ± 0.03	1.13 ± 0.07	5.06 ± 0.05
Soybean	A2704, DP305423,	A2704	0.14 ± 0.0	1.31 ± 0.13	6.03 ± 0.18
	DAS81419, MON89788	DP305423	0.16 ± 0.02	1.50 ± 0.08	6.67 ± 0.20
		DAS81419	0.11 ± 0.01	1.10 ± 0.08	4.74 ± 0.20
		MON89788	0.13 ± 0.03	1.22 ± 0.30	5.00 ± 0.17

^a^ Average of three replications ± standard deviation.

## Data Availability

Data is contained within the article or Appendix A.

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
