# Peer review of "Increasing the Efficiency of Canola and Soybean GMO Detection and Quantification Using Multiplex Droplet Digital PCR"

_biology, 2022, doi:10.3390/biology11020201_

Round 1
Reviewer 1 Report
The reviewed manuscript describes the multiplex ddPCR-based assay for detecting of genetic modifications in canola and soy bean. In the manuscript, the authors described optimization and performance of duplex, triplex, and tetraplex assays. The presented method is one the first developed so far multiplex ddPCR-based technique, which reduces analysis time and allows for mass-scaling of testing. Thus, the content of the manuscript is timely and will be suitable for practical application. However, several comments should be made concerning the study’s design and methodological issues.
Major issues:
- Introduction section – more information about the object is needed, explaining why canola and soy-bean were chosen, as well as why specific GM events were targeted for ddPCR. Currently, there is no information at the section about frequency and abundance of the studied GM plants; the exact reason behind GM-testing also is not stated. Advantages of ddPCR and multiplexing are needed to be described more. Some important citations are also missing.
- Page 2, line 79 – concentration of genomic DNA is not indicated in all experiments. These values are crucial, because amount of DNA affects sensitivity and linearity of ddPCR.
- 1. Section in Results. Optimization of probe concentrations. In accordance to Bio-Rad manual, standard concentrations of primers and probes for ddPCR are 900/250 nM. In supplementary tables other values are given, and the ratio between primers and probes is also different from the recommended one. Extensive description of chosen concentrations is necessary. The reasons behind design of multiplexes are also not clarified, as the reference gene is usually included in multiplex, when in the article references are separated from target genes. There is no explanation, why two different strategies for multiplexing were applied: amplitude multiplexes and probe mixes.
- There are no experiments describing linearity, effects of various DNA concentrations, and sensitivity of the developed assays. No limit of detection and limit of quantification are established.
Minor issues:
- Page 2 line 13 – using of standard curves for real-time PCR. While real time PCR is essentially relative method of quantification, it needs to be clarified why standard curves are necessary for GM detection.
- Page 3, line 86-87. “Three replications of FatA(A) and Lectin assays were run for each GMO mix (0.1, 1 and 5%), which is a total of nine reference runs per assay” – the sentence is unclear and needs further description.
- Page 3, Table 1 – no sequences for reference primers and probes.
- Page 4, lines 147-148. Despite duplexing, two reactions are still needed for the detection of GM as a reference gene is quantified in a separate reaction.
- Page 7, lines 204-205. Optimization of probe concentrations is mentioned but not explained.
Reviewer 2 Report
Dear authors,
This is an interesting study about the feasibility of quantification of two, three and four GM canola and soybean events at the same time using the QX200 ddPCR system.
The topic presents interest, because the developed multiplex ddPCR assays can help to enhance efficiency of GM detection and quantification for canola and soybean events. The abstract abides by all the editing instructions and presents the objectives of the study. The used methods are adequately described. Generally, this is a good work. However, for such a current topic, the references part is very limited. In my opinion, you can improve the quality of your work by adding more references in the Introduction and Discussion sections. The latest articles published on GM detection and quantification using dPCR should be introduced and correlated with the results obtained in this study. For instance, I strongly suggest the following reference:
Detection and Quantification of Genetically Modified Soybean in Some Food and Feed Products. A Case Study on Products Available on Romanian Market. Sustainability 2018, 10(5): 1-13.
Reviewer 3 Report
The main aim of the paper assesses the feasibility of several GM events at the same time with PCR
The study shows interesting information. The introduction should give for information about GM, why it is important its detection and other ways to detect them. Also, it should be remark that information of the genetic modification is needed if CRISPR is used, while other kind of modifications some other approaches can be used. The method and results need improvements, and the discussion should make more emphasis in the importance of the test.
Specific comments
Line 38. Reference needed.
Line 40. Reference needed.
Line 41. Reference needed.
Line 44. Reference needed.
Line 52. More detailed explanation of how is done is needed and the steps to do it.
Line 61. More information of the varieties used is needed. What kind of modification? Where are cultivated?
Line 79. This part needs more explanation. When is duplex or triplex samples? Which proportion of each GM DNA? No control without GM DNA?
Line 148. Which is the threshold limit? Any sample failed to detect GM DNA?
Line 149. Explanation of 0.1%, 1%, 5% what means should be given in the table.
Line 168. The quality of the images is poor. It should be improved.
Line 179. Exceptions? Which ones? They can affect the effectiveness of the test?
Line 237. A more detailed discussion is needed. In an unknown sample will it work. How can authorities use this process?
Round 2
Reviewer 1 Report
The authors of the manuscript have edited the manuscript, and most issues are solved. Only two minor questions are left to be addressed.
- 1 Section in Results: Including figures with the results of primers and probes optimization would improve the quality of the manuscript and be helpful for other studies in the field as it is an essential step in multiplex ddPCR design.
It needs to be specified why 3 different multiplexes have been developed as the same GM events were targets of them all. Reference gene could be included in duplex and triplex assay and save reactions to analyze the single specimen. At the same time, tetraplex assay allows detecting 4 GM events in a single reaction while overall DNA concentration could be determined with a separate reaction with a reference gene. Also, it needs to be clarified whether selected reference genes can be duplicated or not as it could affect DNA quantification. Notably, after proper optimization, tetraplex assay could replace both duplex and triplex. Perhaps, duplex and triplex assays could be considered as steps of optimization for the tetraplex test.
Reviewer 3 Report
Improvements have been made
Author Response
Dear Reviewer,
Thanks for the positive comments.
Thanks